# Unifying physical systems' inductive biases in neural ODE using dynamics constraints

**Yi Heng Lim**  *yi.heng@machine-discovery.com*
*Machine Discovery Ltd.*
*Oxford, United Kingdom*

**Muhammad Firmansyah Kasim**  *authors@machine-discovery.com*
*Machine Discovery Ltd.*
*Oxford, United Kingdom*

**Reviewed on OpenReview:** *https://openreview.net/forum?id=ZOAb497iaY*

## Abstract

Conservation of energy is at the core of many physical phenomena and dynamical systems. There have been a significant number of works in the past few years aimed at predicting the trajectory of motion of dynamical systems using neural networks while adhering to the law of conservation of energy. Most of these works are inspired by classical mechanics such as Hamiltonian and Lagrangian mechanics as well as Neural Ordinary Differential Equations. While these works have been shown to work well in specific domains respectively, there is a lack of a unifying method that is more generally applicable without requiring significant changes to the neural network architectures. In this work, we aim to address this issue by providing a simple method that could be applied to not just energy-conserving systems, but also dissipative systems, by including a different inductive bias in different cases in the form of a regularisation term in the loss function. The proposed method does not require changing the neural network architecture and could form the basis to validate a novel idea, therefore showing promises to accelerate research in this direction.

## 1 Introduction

Learning systems' dynamics with neural networks from data is one of the promising applications of deep neural networks in physical sciences. It has been shown that given enough data, neural networks can uncover the hidden dynamics of physical systems (Chen et al., 2018; Greydanus et al., 2019; Cranmer et al., 2020; Chen et al., 2019) and can be used to simulate the systems with new conditions (Greydanus & Sosanya, 2022).

The most straightforward way in learning a system's dynamics is by training a neural network to learn the system's dynamics given the states at the time, $\mathbf{s}(t)$, i.e.

$$\dot{\mathbf{s}} = \frac{d\mathbf{s}}{dt} = \mathbf{f}(\mathbf{s}(t)). \tag{1}$$

Once the neural network is trained, the system's behaviour can be simulated by solving the equation above using ordinary differential equations solver (ODE). This method is typically called neural ordinary differential equations (NODE) (Chen et al., 2018) in the literature.

Despite its simplicity, NODE has a difficulty in incorporating inductive biases of systems into training. For example, if a system is known to conserve energy, the NODE approach above typically results in growing or shrinking of the energy over a long period of time. A popular workaround is to learn the

Hamiltonian (Greydanus et al., 2019; Toth et al., 2019; Sanchez-Gonzalez et al., 2019) or Lagrangian (Cranmer et al., 2020) of the energy-conserving system with a neural network, then get the dynamics as the derivatives of the learned quantity. Although this approach works in learning energy-conserving systems, it has been shown that to incorporate a specific inductive bias, it takes a significant effort to design a special neural network architecture or to obtain states in special coordinate systems.

Another example is incorporating a bias if the system is known to be dissipative or losing energy. Like in the energy-conserving case, a lot of efforts go into designing special neural network architectures or choosing the optimal weights to ensure stability (Tuor et al., 2020; Xiong et al., 2021). Although those methods work well in getting stable dynamics, changing neural network architectures could potentially reduce the expressiveness of the neural networks.

In this paper, we show how to incorporate various inductive biases in learning systems' dynamics by having various constraints on the dynamics function, including Hamiltonian systems and dissipative systems, without changing the architecture of the neural networks. Where necessary, we also include Invertible Neural Network to transform the coordinate system of the input states, and enforce the inductive biases in the transformed coordinate system. Our code for this paper is available at https://anonymous.4open.science/r/constr-F2D4/.

Our paper makes two main contributions in the form of two distinct regularisation terms for two types of systems, namely energy-conserving and dissipative systems:

1. Derivation of Equation 4 for energy-conserving systems, which is a novel regularisation term that incorporates physical constraints into the training of neural networks for such systems.

2. Derivation of Equation 8 for dissipative systems, based on the argument that such systems are asymptotically stable. This is another novel regularisation term that improves the physical consistency of neural network models for dissipative systems.

## 2  Related works

Motivated by Hamiltonian mechanics, Hamiltonian Neural Network (HNN) was proposed in (Greydanus et al., 2019). HNN takes a set of positions and their corresponding general momenta as input, and models a scalar function called the Hamiltonian. The derivatives of the learned Hamiltonian with respect to the inputs are then computed and multiplied by a symplectic matrix, giving the learned state derivatives. Since the symplectic matrix is manually introduced, HNN has a nice property of observing symplectic structure through time, thus conserving energy. In practice, however, the applicability of HNN is limited by the fact that it requires input data to be prepared in canonical form, which is usually difficult to obtain. There have been a lot of follow-up works that build on the foundation of HNN and expand it to different settings including dissipative systems (Greydanus & Sosanya, 2022; Zhong et al., 2020), generative networks (Toth et al., 2019) and graph networks (Sanchez-Gonzalez et al., 2019), while others try to improve HNN by simplifying it (Finzi et al., 2020; Gruver et al., 2022).

To tackle the shortcoming of HNN, another line of work based on Lagrangian mechanics, called Lagrangian Neural Network (LNN), was proposed in (Cranmer et al., 2020). As the name suggests, the Lagrangian is modelled by a neural network, and the expression to compute the second-order state derivatives could be obtained by algebraically rearranging the Euler-Lagrange equation. With the introduction of Lagrangian prior in the neural network, LNN manages to conserve the total energy of a system without limiting the states to be strictly canonical. Since LNN assumes a coordinate system in which states are easily measured in practice, it can be applied to a wider range of problems where HNN fails. However, to obtain the second-order state derivatives, the inverse Hessian of the neural network must be computed, making LNN computationally expensive, susceptible to ill-conditioning, and sensitive to a poor choice of activation function such as ReLU.

Neural Symplectic Form (NSF) (Chen et al., 2021) exploits the coordinate-free formulation of the Hamiltonian with symplectic 2-form. To speed up the computation, an elegant method to model the symplectic 2-form as an exterior derivative of a parameterised 1-form was proposed. However, there are a few drawbacks in this work. Firstly, without a good understanding of exterior calculus and differential geometry, it could be difficult to understand the motivation behind the work. Secondly, it is expensive to compute the inverse of

the learnable skew-symmetric matrix which requires $\mathcal{O}(N^3)$ operations, where $2N$ is the number of state variables. Thirdly, the training of NSF is unstable and tends to plateau very quickly, see Appendix A.1. To the best of our knowledge, the latter has not been addressed in any work to date.

Invertible Neural Networks (INNs) are neural networks that are elegantly designed in such a way that the inverses of the neural networks are always available with just a forward pass. They have found wide adoption in flow-based generative models and are typically used to map data to a latent space with a simpler distribution. (Dinh et al., 2014; 2016) propose a simple building block called an affine coupling layer whose inverse is easily attainable, and an invertible neural network is obtained by stacking a few of these affine coupling layers in a sequence. In each affine coupling layer, the inputs are permuted since the INNs are not permutation invariant. (Kingma & Dhariwal, 2018) further improves the model by introducing a learnable $1 \times 1$ invertible convolution to replace the fixed permutation of channel dimension used in (Dinh et al., 2016). In connection with NODE, (Zhi et al., 2021) leverages INNs to map the underlying vector field of an ODE system to a base vector field, and others (Chen et al., 2022; Jin et al., 2022) have attempted to include INNs in Hamiltonian systems.

## 3 Methods

Consider a system with states $\mathbf{s} \in \mathbb{R}^{n_s}$, where its states' dynamics depend on the states at the given time, as written in equation 1. The function $\mathbf{f}$ from equation 1 can be represented with an ordinary neural network. The training can then be done by minimizing the loss function,

$$\mathcal{L} = \left\| \hat{\dot{\mathbf{s}}} - \dot{\mathbf{s}} \right\|^2 + w_c \mathcal{C}\left(\dot{\mathbf{s}}\right). \tag{2}$$

The term $\hat{\dot{\mathbf{s}}}$ is the time-derivative of the observed states $\mathbf{s}$ from the experimental data, $\dot{\mathbf{s}}$ is the computed states' dynamics from equation 1, $w_c$ is the constraint's weight, and $\mathcal{C}(\cdot)$ is the constraint applied to the dynamics function, $\dot{\mathbf{s}}$. The last term, i.e. the constraint, is the term that can be adjusted based on the inductive bias of the system.

### 3.1 Hamiltonian systems

Hamiltonian mechanics is a framework to describe particles' equations of motion using an equation in the phase space of the position $q$ and the generalised momenta $p$. This is in contrast to Newtonian's mechanics where it uses forces to describe the equations of motion. Using the Hamilton's equation, $H(q, p)$, a set of equations of motion can be obtained by

$$\frac{d\mathbf{q}}{dt} = \frac{\partial H}{\partial \mathbf{p}}, \ \frac{d\mathbf{p}}{dt} = -\frac{\partial H}{\partial \mathbf{q}}$$

By defining the state space vector as $\mathbf{s} = (\mathbf{q}^T, \mathbf{p}^T)^T$, the states' dynamics of Hamiltonian systems can be written as (Greydanus et al., 2019; Chen et al., 2021)

$$\dot{\mathbf{s}} = \mathbf{J}\nabla H(\mathbf{s}) \ \text{ where } \ \mathbf{J} = \begin{pmatrix} \mathbf{0} & \mathbf{I} \\ -\mathbf{I} & \mathbf{0} \end{pmatrix} \tag{3}$$

where $H$ is the Hamiltonian of the system (typically the energy in some cases) and $\nabla H$ is the gradient of the Hamiltonian with respect to $\mathbf{s}$. The dynamics in the equation above preserve the quantity $H$, i.e. $dH/dt = 0$, which makes it suitable to simulate systems with constant Hamiltonian. If the dynamics are learned using Hamiltonian neural network (Greydanus et al., 2019), the dynamics will be equal to equation 3. The Jacobian in the Hamiltonian system can be written as $\partial\dot{\mathbf{s}}/\partial\mathbf{s} = \mathbf{J}(\partial^2 H/\partial\mathbf{s}\partial\mathbf{s})$. As the Hessian of the Hamiltonian $(\partial^2 H/\partial\mathbf{s}\partial\mathbf{s})$ is a symmetric matrix, the Jacobian of a Hamiltonian system is a Hamiltonian matrix.

Being a Hamiltonian matrix, the Jacobian of the dynamics in equation 3 has special properties. One of them is having zero trace $(\nabla \cdot \dot{\mathbf{s}} = 0)$, which means that there is no source or sink in the vector field of $\dot{\mathbf{s}}$.

Another property is near the local minimum of $H(\mathbf{s})$ (if any), the eigenvalues of the Jacobian $\partial\dot{\mathbf{s}}/\partial\mathbf{s}$ are pure-imaginary which guarantees stability (see appendix A.2 for the derivations).

In order to get the dynamics as in equation 3 by learning the dynamics $\dot{\mathbf{s}}$ directly, it has to be constrained so that the matrix $\mathbf{J}^{-1}(\partial\dot{\mathbf{s}}/\partial\mathbf{s})$ is symmetric. Therefore, we can apply the constraint

$$\mathcal{C}_H\left(\dot{\mathbf{s}}\right) = \left\| \mathbf{J}^T\left(\frac{\partial\dot{\mathbf{s}}}{\partial\mathbf{s}}\right) - \left(\frac{\partial\dot{\mathbf{s}}}{\partial\mathbf{s}}\right)^T \mathbf{J} \right\|_F^2, \tag{4}$$

where $\|\cdot\|_F$ is the Frobenius norm to get the matrix $\mathbf{J}^{-1}(\partial\dot{\mathbf{s}}/\partial\mathbf{s})$ as symmetric as possible. Note that $\mathbf{J}^{-1} = \mathbf{J}^T$.

**Theorem 3.1.** *The dynamics of the states, $\dot{\mathbf{s}} \in \mathbb{R}^n$, follow the form in equation 3 for a function $H(\mathbf{s}) : \mathbb{R}^n \to \mathbb{R}$ if and only if the matrix $\mathbf{J}^{-1}(\partial\dot{\mathbf{s}}/\partial\mathbf{s})$ is symmetric.*

*Proof.* If the dynamics $\dot{\mathbf{s}}$ follow equation 3, then the matrix $\mathbf{J}^{-1}(\partial\dot{\mathbf{s}}/\partial\mathbf{s})$ represents the Hessian of the Hamiltonian $H$, which is symmetric. To prove the converse, note that the matrix $\mathbf{J}^{-1}(\partial\dot{\mathbf{s}}/\partial\mathbf{s}) = (\partial\dot{\mathbf{z}}/\partial\mathbf{s})$ can be a Jacobian matrix of $\mathbf{p} = (-\dot{\mathbf{s}}_l, \dot{\mathbf{s}}_f)$ where $\dot{\mathbf{s}}_f$ and $\dot{\mathbf{s}}_l$ are the first half and the last half elements of $\dot{\mathbf{s}}$. If the matrix $(\partial\mathbf{p}/\partial\mathbf{s})$ is a symmetric Jacobian matrix, then the expression below must be evaluated to zero,

$$\sum_{i>j}\left(\frac{\partial p_i}{\partial s_j} - \frac{\partial p_j}{\partial s_i}\right)\mathrm{d}s_i \wedge \mathrm{d}s_j = \sum_{i,j}\left(\frac{\partial p_i}{\partial s_j}\right)\mathrm{d}s_i \wedge \mathrm{d}s_j = \mathrm{d}f = 0,$$

where $f = \sum_i p_i ds_i$. As $\mathrm{d}f = 0$, by Poincaré's lemma, there exists an $\alpha$ such that $\mathrm{d}\alpha = f$. In other words, if the matrix $(\partial\mathbf{p}/\partial\mathbf{s})$ is a symmetric Jacobian matrix, then there exists an $\alpha$ such that its Hessian is equal to the Jacobian matrix, i.e. $(\partial^2\alpha/\partial\mathbf{s}\partial\mathbf{s}) = (\partial\mathbf{p}/\partial\mathbf{s})$. This $\alpha$ corresponds to the Hamiltonian $H$ in equation 3. $\square$

### 3.2 Coordinate-transformed Hamiltonian systems

One of the drawbacks of Hamiltonian systems is that they have to use the right coordinates for the states, usually known as the canonical coordinates. In simple cases like simple harmonic motion of a mass-spring system, the canonical coordinates can easily be found (i.e. they are the position and momentum of the mass). However, in more complicated cases, finding the canonical coordinates for Hamiltonian sometimes requires expertise and analytical trial-and-error.

Let us denote the canonical states' coordinates as $\mathbf{z}$ and the observed states' coordinates as $\mathbf{s}$. As the canonical states' coordinates $\mathbf{z}$ are not usually known, we can find them using a learnable invertible transformation,

$$\mathbf{z} = \mathbf{g}(\mathbf{s}) \quad \text{and} \quad \mathbf{s} = \mathbf{g}^{-1}(\mathbf{z}), \tag{5}$$

then learn the dynamics of $\mathbf{z}$ with a neural network, i.e. $\dot{\mathbf{z}} = \mathbf{f}_{\mathbf{z}}(\mathbf{z}, t)$. This way, the dynamics of $\mathbf{s}$ can be computed by $\dot{\mathbf{s}} = (\partial\mathbf{g}^{-1}/\partial\mathbf{z})\dot{\mathbf{z}}$. The invertible transformation above can be implemented using an invertible neural network (Kingma & Dhariwal, 2018). With the invertible transformation between $\mathbf{s}$ and $\mathbf{z}$, the dynamics of $\mathbf{s}$ in this case can be written as,

$$\dot{\mathbf{s}} = \mathbf{G}^{-1}(\mathbf{s})\mathbf{J}\mathbf{G}^{-T}(\mathbf{s})\nabla H'(\mathbf{s}), \tag{6}$$

where $H'(\mathbf{s}) = H\left(\mathbf{g}(\mathbf{s})\right)$, and matrix $\mathbf{G}(\mathbf{s}) = \partial\mathbf{g}/\partial\mathbf{s}$ is the Jacobian of the transformation function from $\mathbf{s}$ to $\mathbf{z}$ from equation 5. It can also be shown that the dynamics in equation 6 preserve the quantity of $H'$.

**Theorem 3.2.** *For $\dot{\mathbf{s}}$ following the equation 6, the quantity of $H'(\mathbf{s})$ are constant throughout the time, i.e. $dH'(\mathbf{s})/dt = 0$.*

*Proof.* Denote the matrix $\mathbf{D}$ as $\mathbf{D} = \mathbf{G}^{-1}\mathbf{J}\mathbf{G}^{-T}$, where the matrix $\mathbf{D}$ is a skew-symmetric, because

$$\mathbf{D}^T = (\mathbf{G}^{-1}\mathbf{J}\mathbf{G}^{-T})^T = \mathbf{G}^{-1}\mathbf{J}^T\mathbf{G}^{-T} = -(\mathbf{G}^{-1}\mathbf{J}\mathbf{G}^{-T}) = -\mathbf{D}.$$

The rate of change of $H'$ can be written as $dH'(\mathbf{s})/dt = \mathbf{u}^T\dot{\mathbf{s}} = \mathbf{u}^T\mathbf{D}\mathbf{u}$, where $\mathbf{u} = \nabla H'(s)$. As the quantity above is a scalar, it must be equal to its transpose. Thus,

$$\mathbf{u}^T\mathbf{D}\mathbf{u} = \left[\mathbf{u}^T\mathbf{D}\mathbf{u}\right]^T = \mathbf{u}^T\mathbf{D}^T\mathbf{u} = -\mathbf{u}^T\mathbf{D}\mathbf{u} = 0$$

as $\mathbf{D}^T = -\mathbf{D}$ because of its skew-symmetricity. Therefore, $dH'(\mathbf{s})/dt = 0$. □

As this system is similar to the Hamiltonian system in the previous subsection, we can apply the constraint below,

$$\mathcal{C}_{H'}(\dot{\mathbf{s}}) = \mathcal{C}_H(\dot{\mathbf{z}}) \tag{7}$$

where $\mathcal{C}_H(\cdot)$ is the constraint from equation 4.

### 3.3 Dissipative systems

Dissipative systems have a characteristic that their dynamics are asymptotically stable due to the frictions applied to the systems. In terms of ODE dynamics, this asymptotic stablility can be achieved by having the real parts of all eigenvalues of the Jacobian $\partial\dot{\mathbf{s}}/\partial\mathbf{s}$ to be negative. This would ensure stability around the stationary point. Therefore, the following constraint can be applied to enforce the condition,

$$\mathcal{C}_D(\dot{\mathbf{s}}) = \sum_i \left| \max\left\{0, \mathrm{Re}\left[\lambda_i\left(\frac{\partial\dot{\mathbf{s}}}{\partial\mathbf{s}}\right)\right] - a_i\right\}\right|^2, \tag{8}$$

where $\lambda_i(\cdot)$'s are the eigenvalues of the Jacobian $\partial\dot{\mathbf{s}}/\partial\mathbf{s}$, $a_i$'s are the assumed upper bounds of the real parts of the eigenvalues. The upper bounds $a_i$'s are sometimes useful in making sure that the eigenvalues are far from 0, further ensuring the stability.

## 4 Experiments

In order to see if the methods described above work, we tested them on various physical systems.

**Task 1 - Ideal mass-spring system.** The first case is an ideal mass-spring system. The Hamiltonian canonical state of this system is $(x, m\dot{x})^T$ where $m$ is the mass and $x$ is the displacement of the mass from its equilibrium position. The acceleration is given by $-(k/m)\ddot{x}$. In our experiments, we set $m = 1$ and the spring constant $k = 1$, and the canonical state is simply $(x, \dot{x})^T$. We generated the data by randomly initialising the state from a normal distribution, and added white noise with $\sigma = 0.1$ to the state trajectory. For training, we generated 250 trajectories, with 30 samples in each trajectory equally spaced within $t = [0, 2\pi]$.

**Task 2 - Ideal rod-pendulum.** The second case is an ideal rod-pendulum. The Hamiltonian canonical state of this system is $(\theta, m\dot{\theta})^T$ where $m$ is the mass and $\theta$ is the angular displacement of the mass from its equilibrium position. In our experiment, we set $m = 1$, and the canonical state becomes $(x, \dot{x})^T$. The angular acceleration is given by $\ddot{\theta} = -(g/l)\sin\theta$ with the value of $g$ and $l$ set to 3 and 1 respectively. Similar to Task 1, we generated the data by randomly initialising the state from a normal distribution, and added white noise with $\sigma = 0.1$ to the state trajectory. For training, again, we generated 250 trajectories, with 30 samples in each trajectory equally spaced within $t = [0, 2\pi]$. This task is slightly more complicated than Task 1 since it is a non-linear dynamical system.

**Task 3 - Ideal double rod-pendulum.** The third case is an ideal double rod-pendulum. We chose a general state of $(\theta_1, \theta_2, \dot{\theta}_1, \dot{\theta}_2)^T$ since the Hamiltonian canonical state of this problem is non-trivial to obtain. Again, we set the the two masses $m_1 = m_2 = 1$ and the two rod lengths $l_1 = l_2 = 1$, and generated the data by randomly initialising the state from a normal distribution and obtained the trajectory. With the aforementioned parameters, the two angular accelerations are given by

$$\ddot{\theta}_1 = \frac{3\sin(\theta_1) - \sin(\theta_1 - 2\theta_2) - 2\sin(\theta_1 - 2\theta_2)(\dot{\theta}_2^2 + \dot{\theta}_1^2)\cos(\theta_1 - \theta_2)}{3 - \cos(2\theta_1 - 2\theta_2)}$$

$$\ddot{\theta}_2 = \frac{2\sin(\theta_1 - \theta_2)(2\dot{\theta}_1^2 - 2\cos(\theta_1)\dot{\theta}_2^2\cos(\theta_1 - \theta_2))}{3 - \cos(2\theta_1 - 2\theta_2)}$$

Since this is a chaotic dynamical system, we omit the addition of noise to the trajectory to avoid introducing irreducible aleatoric uncertainty, and generated more data for training: 2000 trajectories, with 300 samples in each trajectory equally spaced within $t = [0, 2\pi]$.

**Task 4 - Damped single rod-pendulum with redundant states.** In this case the pendulum is damped with coefficient $\alpha$ to model a dissipative system. The angular acceleration is given by $\ddot{\theta} = -(g/l)\sin\theta - \alpha\dot{\theta}$ with $m = l = g = 1$ and $\alpha = 0.05$. The observed states are $(x, y, \dot{x}, \dot{y})^T$ where $x$ and $y$ are respectively the horizontal and vertical positions from the pivot, i.e. $x = l\sin\theta$ and $y = -l\cos\theta$. These states are selected instead of $(\theta, \dot{\theta})$ to illustrate a system where the observed states are redundant. As this is a dissipative system, we use the constraint from equation 8 for the training.

## 4.1 Training details

For Task 1 and Task 2, we define our method as being a simple Neural Ordinary Differential Equation (NODE), but with constraint 4 added to the loss function. For Task 3, since the input states were not in canonical form, we added a component of Invertible Neural Network (INN), with the FrEIA[1] package, based on the invertible architecture defined in (Kingma & Dhariwal, 2018) to transform the state coordinate system in our method, and applied constraint 7. For Task 4, since it is a dissipative system, we added constraint 8 to the loss function.

In the first three tasks, we compared our method with baseline NODE, HNN, LNN and NSF. For Task 4, since it is well established that HNN, LNN and NSF would not work in dissipative systems, we only compared our method to baseline NODE. We trained all models in all tasks with Adam optimiser (Kingma & Ba, 2014) and the same learning rate of $10^{-4}$. We constructed multi-layer perceptrons (MLPs) for all models, with 3 layers of 200 hidden units in each case. INN with 8 blocks was used where needed, with each block having 2 layers of 100 hidden units. We chose softplus as the activation function for all models as it worked well in all cases.

In our experiments, LNN and NSF were difficult to train and training inconsistencies were often incurred by these two models. The training inconsistencies of LNN were pointed out in (Chen et al., 2021) and we showed the inconsistencies of NSF in Appendix A.1. In those cases, we ran the experiments several times to obtain the best training results so that we could compare our method to the best trained models. We found that using a smaller batch size to increase the number of training steps helped in reducing those inconsistencies. We note that this is a weakness of LNN and NSF as the inconsistencies might make training difficult in practical applications.

For our method, we set the coefficient of the constraint as large as possible without compromising the loss of time derivatives of states. In Task 1, 2, 3 and 4 respectively, we set the coefficient to be $10^5$, $10^4$, $10^3$ and $10^2$. We used a batch size of 32 for Task 1, Task 2 and Task 4, and a larger batch size of 1280 for Task 3 since there were more data. We ran all experiments for 1000 epochs on a single NVIDIA Tesla T4 GPU.

To test, we started from a randomly generated state at time 0, and rolled out the subsequent states from time 0 to 100, and logged the RMSE of the energy deviations. Note that the initial state at testing is different from the training data. We recorded the median, 2.5[th] percentile and 97.5[th] percentile since there are outliers in some of the test results.

## 5 Results

Overall, our method performed consistently well and ranked in the top two methods in all tasks in terms of energy drift from the ground truth. The other methods returned mixed results, performing well in some tasks but not the others. While NSF performed reasonably well in all tasks, we show in Appendix A.1 that training inconsistencies are often encountered in NSF.

---

[1]https://github.com/VLL-HD/FrEIA

| Case | Task 1 | Task 2 | Task 3 | Task 4 |
|------|--------|--------|--------|--------|
| NODE | $0.086^{+0.095}_{-0.077}$ | $\infty$ | $\infty$ | $0.18^{+718.10}_{-0.059}$ |
| HNN | $0.0065^{+0.013}_{-0.0045}$ | $\mathbf{0.015^{+0.066}_{-0.012}}$ | $24.44^{+8.69}_{-11.40}$ | N/A |
| LNN | $0.0060^{+0.021}_{-0.0046}$ | $0.032^{+0.10}_{-0.022}$ | $7.92^{+124.44}_{-7.62}$ | N/A |
| NSF | $0.014^{+0.30}_{-0.0093}$ | $0.027^{+0.067}_{-0.016}$ | $4.29^{+15.07}_{-3.05}$ | N/A |
| Our method | $\mathbf{0.0017^{+0.0069}_{-0.0012}}$ | $0.024^{+0.068}_{-0.013}$ | $\mathbf{4.03^{+10.52}_{-3.16}}$ | $\mathbf{0.12^{+0.39}_{-0.012}}$ |

Table 1: RMSE of the energy deviations of 100 test trajectories. The numbers shown are mean $\pm$ standard deviation. Since there are outliers in some of the results, the median, $2.5^{\text{th}}$ and $97.5^{\text{th}}$ of the RMSE in each case are logged. The best result in each case, judging by the $97.5^{\text{th}}$ of the RMSE, is bolded, and the second best result is highlighted in blue. $\infty$ indicates numerical overflow.

Our method performed the best in Task 1 and was on par with HNN in Task 2 which logged the best performance. This is consistent with our expectation since the input coordinates are in canonical form. It is worth noting that even though the symplectic structure is enforced inherently in the HNN architecture while our method employed a soft constraint, our method still compared competitively with HNN in the first two tasks. More importantly, as can be seen in Figure 1(a) and (b), similar to the other known energy-conserving models, the periodic drift in energy of our model oscillates around a constant offset from the $x$-axis throughout the time span. While this periodic drift is mostly attributed to the state prediction error due to the white noise added to the training data, the constant offset from the $x$-axis means that the average perceived energy in the system is conserved.

A more interesting point to note is that our method outperformed HNN in Task 1, even though in both models the symplectic structure is preserved. We suspect this is because the inherent symplecticity in HNN acts as a double-edged sword: although it is good at preserving the symplecticity, it reduces the flexibility of the model in some cases especially in the presence of noise.

Task 3 is the most difficult system to model among the first three tasks since the motion of a double-pendulum is chaotic and the input coordinates are not in canonical form. Table 1 shows that our method performed the best in terms of energy drift of the predicted system. This is further corroborated by Figure 1(c) where again, the energy drift is at a constant periodic offset from the $x$-axis.

In Task 4, our method performed significantly better than vanilla NODE. It is interesting to note that while NODE failed spectacularly in the edge cases as indicated by the large RMSE upper bound in terms of energy drift, our method worked reasonably well by just including a simple dissipative bias in the loss function. Figure 1(d) shows one of the worse predictions that NODE returned, and the corresponding states are shown in Figure 2 and Figure 3.

The states in Task 4 are redundant as there are 4 states used while there are only 2 true states. This makes the dynamics of the pendulum lie on a low dimensional manifold in a 4-dimensional space. Without the constraint, the error in learning the dynamics by NODE could send the states outside the manifold where no training data exists, thus making the dynamics unstable. On the other hand, the constraint in equation 8 makes sure the dynamics are stable around some stationary points which typically lie on the manifold. Therefore, a slight error in learning the dynamics in our case can be corrected by attracting it back to the stationary point on the manifold. This is an important example of ensuring the stability of a novel system since the true number of states are usually unknown.

# 6 Pixel pendulum

## 6.1 Energy-conserving pendulum

Similar to (Greydanus et al., 2019) and (Chen et al., 2021), we tested our model with pixel data. This resembles our model in Task 3 where INN is used to transform the state coordinates, except that in this case

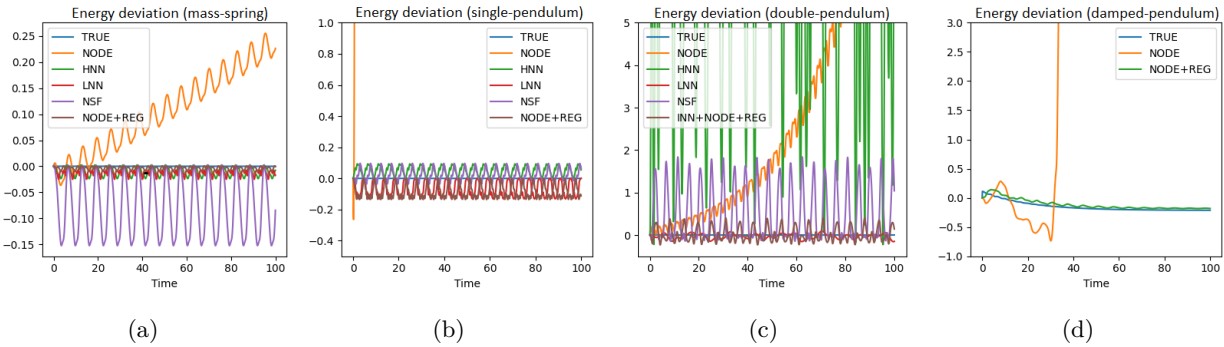

Figure 1: The energy drift from ground truth in every model for (a) mass-spring (Task 1), (b) single-pendulum (Task 2), (c) double-pendulum (Task 3) and (d) damped-pendulum (Task 4). All test cases were taken from the same distribution as the training data, except for (c) where the test case was sampled from $\mathbf{s}_i \sim \mathcal{U}(-0.5, 0.5), \forall i$ for better illustration.

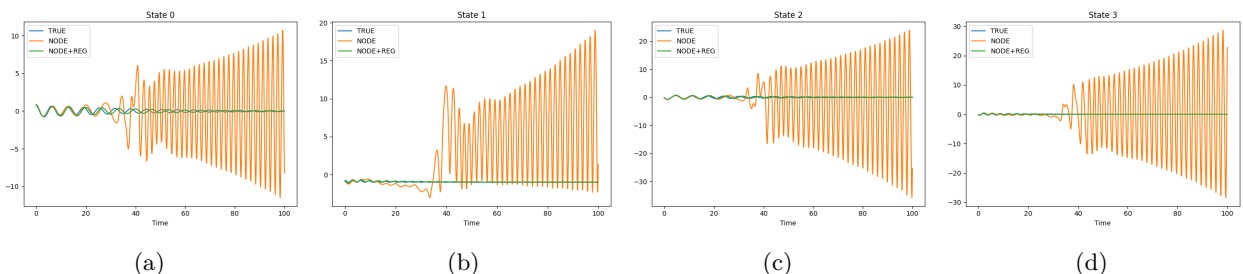

Figure 2: The trajectories of the four states of damped-pendulum in Task 4. This sample is picked randomly from one of the edge cases in NODE. (a) shows the trajectory of $\sin\theta$, (b) shows the trajectory of $-\cos\theta$ while (c) and (d) are the trajectories of the horizontal and vertical velocities respectively.

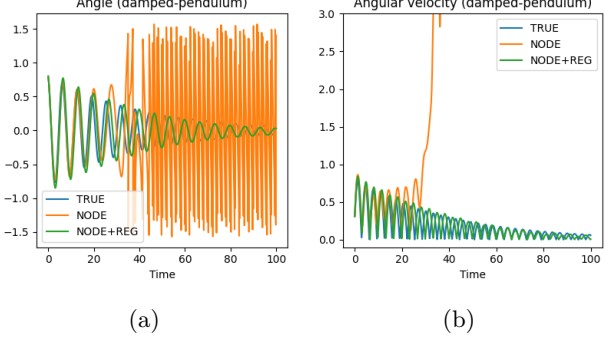

Figure 3: The trajectories of $\theta$ and $\dot{\theta}$ of damped-pendulum in Task 4, transformed from Figure 2.

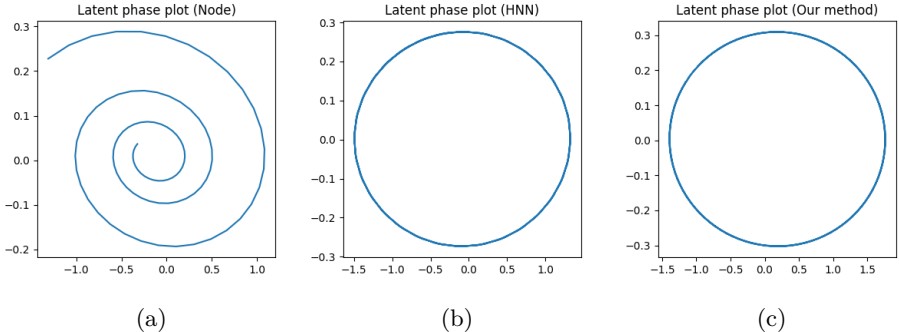

Figure 4: Phase plot of the learned dynamics in the encoded latent space. Trained (a) vanilla NODE, (b) HNN and (c) our method. The learned dynamics here are the learned states, i.e. angular position and angular velocity of the mass, rolled out in time.

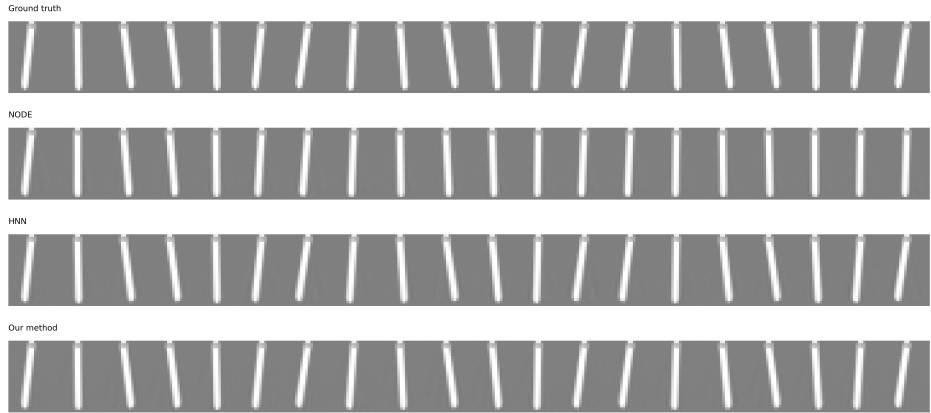

Figure 5: Predictions of the pendulum position with image data from time 0 to 100.

we seek to encode image data in a latent coordinate system. We used the same training hyperparameters as the previous experiments, except that a regularisation weight of 500 is used here for our model and 160 videos of 100 frames equally spaced within $[0, 100]$ each were used as the training data. We also found that ReLU activation function worked better for the autoencoder. Figure 5 shows the predictions for $t = [0, 100]$ using NODE, HNN, and our method while Figure 4 shows the phase plots in the encoded latent space. The phase plot of NODE, as expected, spirals inwards indicating a drift in the symplectic structure, and Figure 5 shows that its predicted pendulum dynamics are barely moving due to the energy drift. On the other hand, HNN and our method performed similarly well with the predicted pendulum dynamics closely resemble the ground truth. The phase plots of both HNN and our method are perfect circles, meaning there is no sink or source in the latent vector field and the symplectic structure is preserved. This is, however, a simple proof of concept with a vanilla autoencoder. We note that going forward, we could improve the results by using a better autoencoder or even a pretrained model.

## 6.2 Damped pendulum

To test our model for dissipative systems with image data, we modified the code from OpenAI pendulum gym (Brockman et al., 2016) to include a damping factor of 0.05 to the pendulum motion. In this case, we set the number of latent states to 20 to mirror the fact that the number of states in a system is usually unknown, and added constraint 8 weighted by 500 to the vanilla NODE in our method. In Figure 6, the predictions of NODE are noisy due to large errors in latent space. As explained in Task 4, these errors are due to the

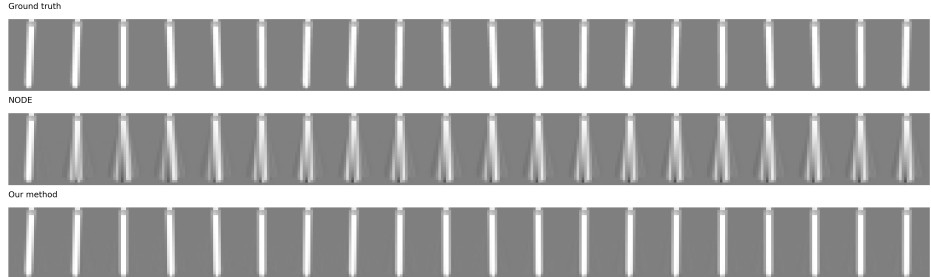

Figure 6: Predictions of the damped pendulum position with image data from time 0 to 100.

predicted states being outside of the 2D manifold. In contrast, with a simple constraint, we successfully reduced these errors and removed the noise from the pixel predictions.

## 7  Limitations

Even though our method is simple and proven to work well in different scenarios including both energy-conserving and dissipative systems, we have introduced a free hyper-parameter which is the regularisation weight. This weight could be difficult to fine-tune depending on the experiments: setting too low a value may not enforce the constraint while setting too high a value may jeopardise the training. This fine-tuning process is further exacerbated if we have more than one regularisation term. As part of the future work, we may explore using Lagrangian multipliers as a potential solution.

## 8  Broader impact

Unlike classical control theory that has a strong theoretical foundation to explain every phenomenon and trajectory in a deterministic dynamical system, the results obtained from neural networks could be inexplicable at times.

## 9  Conclusions

In this paper, we proposed a simple method to enforce inductive bias in a vanilla Neural Ordinary Differential Equation (NODE) to model both energy-conserving and dissipative systems. This was conducted by incorporating a regularisation term in the loss function: in energy-conserving systems the Hamiltonian symplectic structure is enforced whereas in dissipative systems the state vector fields are made to spiral towards a sink. Our method unifies different approaches in modelling dynamical systems, and is more generally applicable since it does not require significant changes to the neural network architectures, nor does it require data in specific coordinate systems, thus showing promises to accelerate research in this domain.

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

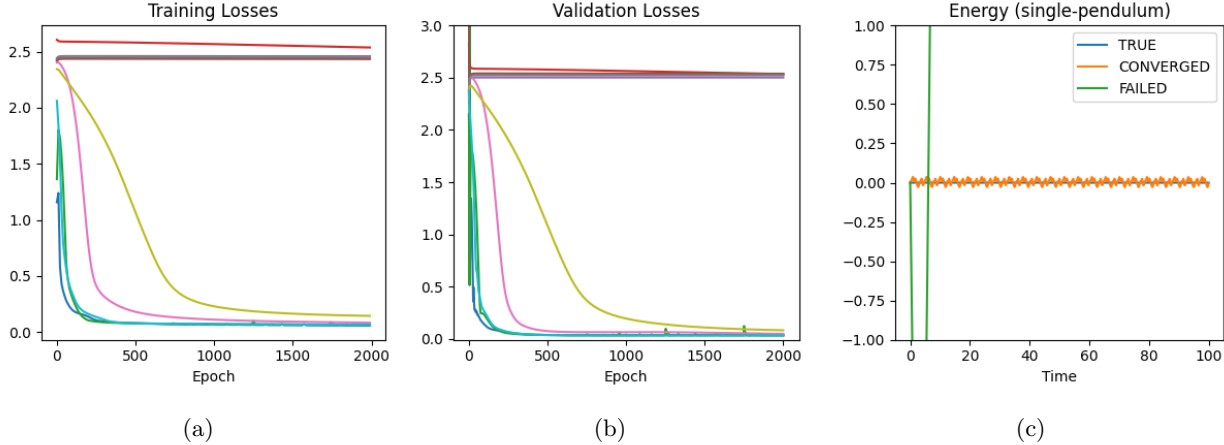

Figure 7: (a) Training losses of 10 identical NSF runs, (b) Validation losses of 10 identical runs, and (c) Comparison of energy conservation between failed and successful NSF runs.

## A Appendix

### A.1 NSF training inconsistency

To show the training inconsistency often incurred by NSF, we repeated the training of Task 2 with NSF 10 times using the same data and plotted the losses of each training, see Figure 7. We used a single batch that included all data, and a learning rate of $10^{-3}$ here. We note that this problem could be mitigated, but not removed, by using a smaller batch size, which we did in the main paper. We also note that this inconsistency is common to all tasks. The early plateaus in some of the runs are not due to the expressivity of the neural networks, since the exact same configuration was used in all runs. Even though NSF, when converged, gives relatively good performance in terms of energy conservation, this training inconsistency is unacceptable in any practical application.

### A.2 Stability of Hamiltonian near a local minimum

The dynamics of Hamiltonian system in equation 3 can be written as

$$\dot{\mathbf{s}} = \mathbf{J}\nabla H(\mathbf{s}) \quad \text{where} \quad \mathbf{J} = \begin{pmatrix} \mathbf{0} & \mathbf{I} \\ -\mathbf{I} & \mathbf{0} \end{pmatrix}. \tag{9}$$

The Jacobian matrix of the dynamics $\partial\dot{\mathbf{s}}/\partial\mathbf{s}$ is then

$$\frac{\partial\dot{\mathbf{s}}}{\partial\mathbf{s}} = \mathbf{J}\mathbf{B} \quad \text{where} \quad \mathbf{B} = \frac{\partial^2 H(\mathbf{s})}{\partial\mathbf{s}\partial\mathbf{s}} \tag{10}$$

where $\mathbf{B} = \partial^2 H(\mathbf{s})/\partial\mathbf{s}\partial\mathbf{s}$ is the symmetric Hessian matrix of the Hamiltonian. Near a local minimum of $H(\mathbf{s})$, the Hessian matrix $\mathbf{B}$ is a positive definite matrix. With positive definitive property of $\mathbf{B}$, there exists the matrix square root and its inverse, i.e., $\mathbf{B}^{1/2}$ and $\mathbf{B}^{-1/2}$ which are also symmetric matrices. Therefore, the matrix $\mathbf{J}\mathbf{B}$ is similar to $\mathbf{B}^{1/2}\mathbf{J}\mathbf{B}^{1/2}$ as

$$\mathbf{B}^{1/2}\mathbf{J}\mathbf{B}^{1/2} = \mathbf{B}^{1/2}(\mathbf{J}\mathbf{B})\mathbf{B}^{-1/2}. \tag{11}$$

As $\mathbf{J}$ is a skew-symmetric matrix and $\mathbf{B}^{1/2}$ is a symmetric matrix, the matrix $\mathbf{B}^{1/2}\mathbf{J}\mathbf{B}^{1/2}$ is a skew-symmetric matrix.

Being a skew-symmetric matrix, the eigenvalues of $\mathbf{B}^{1/2}\mathbf{J}\mathbf{B}^{1/2}$ are pure imaginary. Therefore, from the similarity, we can conclude that the eigenvalues of $\mathbf{J}\mathbf{B}$ are also pure imaginary near the local minimum of $H(\mathbf{s})$.

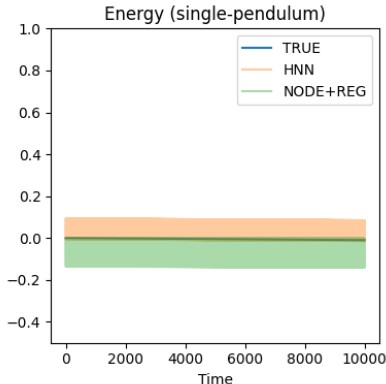

Figure 8: Comparison between HNN and our method in Task 2 when the models are rolled out in a longer time span during inference.

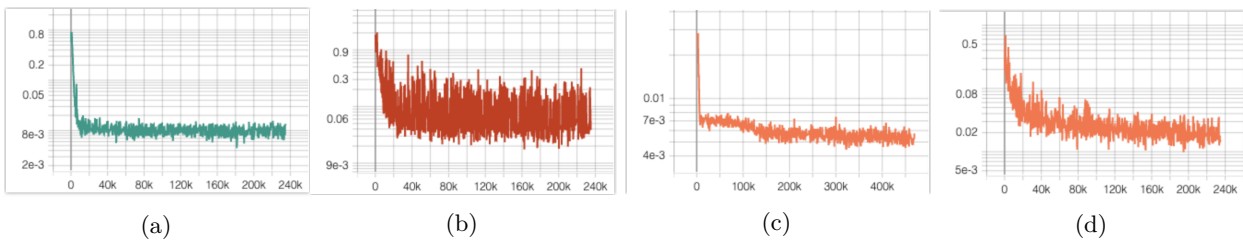

Figure 9: (a) Training losses of our method for (a) mass-spring (Task 1), (b) single-pendulum (Task 2), (c) double-pendulum (Task 3) and (d) damped-pendulum (Task 4). Logarithmic loss is plotted against the number of training steps in each plot.

## A.3 Additional plots

We include Figure 8 to show the reliability of our method during inference when the prediction spans a long time horizon.

We also include Figure 9 to show the training curves of our method for Tasks 1 to 4.

