# OpenReview forum: "Unifying physical systems’ inductive biases in neural ODE using dynamics constraints"
_TMLR — Accepted by TMLR_

### Review · Reviewer_rNfb · 2022-12-29

**Summary Of Contributions:**

Learning the dynamics of a physical system is often difficult because we can't enforce constraints such as conservation of energy. The authors propose adding a regularization term to the loss function that captures physical constraints. They train models on several benchmark problems and compare to several baselines.

**Audience:**

Yes

**Claims And Evidence:**

Yes

**Requested Changes:**


- In Eq. 6, I think there is a `-` sign that is meant to be a `=`.
- For tasks 1-3 the authors clearly explain how the training data was generated. It would be nice if they could add similar descriptions for the test data in tasks 1-3 and for both datasets in task 4. In particular, they should emphasize that the test data are distinct from the training data.
- The caption for Table 1 says it shows mean $\pm$ standard deviation, but the uncertainty is not symmetric. Are these actually the 2.5 and 97.5 percentiles?


**Strengths And Weaknesses:**

# Strengths

- The authors clearly explain the challenge of including physics-based constraints and inductive biases and give a good overview of existing methods.
- The chosen tasks are explained well, and seem like appropriate simple test cases.


# Concerns

- It is a little unclear exactly what theorems and methods are novel contributions. I would suggest adding an itemized list of contributions at the end of the introduction.
- I think the assumed physics knowledge in this paper is going to be a challenge for a large portion of the TMLR audience. I realize that space is at a premium, but a very short introductory example in section 3.1 would be helpful--describing the Hamiltonian and state for an ideal mass-spring for example.
- IIUC the only test metric is squared deviation from ground truth energy, with no consideration given to whether the actual dynamics are correct. This allows for very incorrect solutions to perform well e.g. a solution where the system sits still in tasks 1-3. I would suggest adding some evaluation metrics to reassure the reader that the dynamics are actually correct.
- In the constraint for task 4, there are parameters $a_i$ that place upper bounds on the real parts of the eigenvalues of the Jacobian. How are these values chosen? Don't they determine how dissipative the system is, so choosing them "just right" already gives your network a significant amount of knowledge? How would we go about choosing these for an unknown dissipative system?

---

> ### Author Response · Authors · 2023-02-21
> **Replies to Reviewer rNfb**
>
> > It is a little unclear exactly what theorems and methods are novel contributions. I would suggest adding an itemized list of contributions at the end of the introduction.
>
> We thank the reviewer for the suggestion, and we have added that to the introduction.
>
> > I think the assumed physics knowledge in this paper is going to be a challenge for a large portion of the TMLR audience. I realize that space is at a premium, but a very short introductory example in section 3.1 would be helpful--describing the Hamiltonian and state for an ideal mass-spring for example.
>
> We thank the reviewer again for the suggestion, and we have added a brief example to address that in section 3.1.
>
> > IIUC the only test metric is squared deviation from ground truth energy, with no consideration given to whether the actual dynamics are correct. This allows for very incorrect solutions to perform well e.g. a solution where the system sits still in tasks 1-3. I would suggest adding some evaluation metrics to reassure the reader that the dynamics are actually correct.
>
> In tasks 1-3 of our study, the systems are in motion and have non-zero energy. A model that predicts the system to be stationary (and thus have 0 energy) will have a significant deviation from the ground truth energy. There may have been some confusion regarding Figure 1, where we labelled the axis as "Energy" instead of "Energy deviation". We have since corrected this label in the revised paper.
>
> > In the constraint for task 4, there are parameters that place upper bounds on the real parts of the eigenvalues of the Jacobian. How are these values chosen? Don't they determine how dissipative the system is, so choosing them "just right" already gives your network a significant amount of knowledge? How would we go about choosing these for an unknown dissipative system?
>
> In Equation 8 of the paper, we introduced negative upper bounds to ensure technical correctness. However, in practice, we have found that setting the upper bounds to 0 works well for our model.
>
> > In Eq. 6, I think there is a - sign that is meant to be a =.
>
> It is actually a "=" in the paper.
>
> > For tasks 1-3 the authors clearly explain how the training data was generated. It would be nice if they could add similar descriptions for the test data in tasks 1-3 and for both datasets in task 4. In particular, they should emphasize that the test data are distinct from the training data.
>
> We thank the reviewer for this comment, and have addressed this in the revised paper.
>
> > The caption for Table 1 says it shows mean standard deviation, but the uncertainty is not symmetric. Are these actually the 2.5 and 97.5 percentiles?
>
> Yes, they are the 2.5 and 97.5 percentiles.

---

> > ### Comment · Reviewer_rNfb · 2023-03-13
> > **Thanks for the clarification**
> >
> > >In tasks 1-3 of our study, the systems are in motion and have non-zero energy. A model that predicts the system to be stationary (and thus have 0 energy) will have a significant deviation from the ground truth energy. There may have been some confusion regarding Figure 1, where we labelled the axis as "Energy" instead of "Energy deviation". We have since corrected this label in the revised paper.
> >
> > The confusion comes from the definition of "energy deviation" as an evaluation metric. I interpreted this to mean "deviation from the system's initial energy." When the system is conservative, we would expect this to be zero. However, it is deviation from the energy computed from the training data. Thanks for clearing this up.
> >
> > I'm still a little concerned with it as an evaluation metric though---matching the energy of the ground truth system is necessary but not sufficient for matching the dynamics. It would be nice to see plots that show your method gets the position + velocity correct as well.

---

> > > ### Author Response · Authors · 2023-03-14
> > > **Replies to Reviewer rNfb**
> > >
> > > We thank the reviewer for the follow-up question, and we would like to draw the reviewer's attention to figures 4 to 6, which address the concern in detail. Specifically, Figure 4 shows the phase plot (angular velocity vs angular position) of the encoded latent space when the model is trained with pixel data. Furthermore, Figure 5 and Figure 6 show the positions of the image pendulum rolled out in time.

---

### Review · Reviewer_vbRE · 2023-01-19

**Summary Of Contributions:**

This paper considers the problem of learning physical systems’ dynamics by machine learning models.  It proposes to use the regularization term in loss function to incorporate inductive biases which are often inferred by physics, such as conservation laws. The regularization term penalizes dynamics that are away from the physical laws. The paper also experimentally evaluated the proposed method on several simple physics systems. The performance of the proposed method outperforms several other related methods in most of the examined systems.

**Audience:**

Yes

**Claims And Evidence:**

No

**Requested Changes:**

I think the authors need to evaluate the method in more complicated physics systems, and to show it works well in these scenarios.

I hope the authors can show an efficient way of managing the scenarios of multiple inductive biases and constraints.

I would like the authors to also provide evidence for “it has been shown that to incorporate a specific inductive bias it takes a significant effort ……”

Provide evidence for the claim “changing neural network architectures could potentially reduce the expressiveness of the neural network”.


**Strengths And Weaknesses:**

Strength:

The proposed method of adding a regularization term for inductive bias is easy to implement.

In the experiments presented in the paper, the proposed method has good performance, compared to other similar methods.

Weakness:

> The idea of adding regularization to penalize unwanted aspects of learning is a very common trick in the machine learning community. Hence, the novelty is limited.

> The experiments shown in the paper are only very simple physics systems, which have very simple physics solutions. For a good evaluation of the proposed method, I expect it is tested on more complicated systems.

> I have doubts on the effectiveness of the proposed method. I noticed that all the experiments use super large constraint coefficients so that the enforced constraints are strictly obeyed. According to my personal experience, there is always a tradeoff between learning the system (corresponding to the loss without regularization) and enforcing the regularization. When the constraint coefficient is so large as in this paper, the machine learning models usually learn little to nothing meaningful. Hence, I suspect that the good performance in the presented experiments might be because the physics systems are so simple that they are very easy to learn. If this is the case, the proposed method may not work well for complicated systems.

> Another doubt I have is that: when there exist several inductive biases (for example several conservation laws), there should be multiple regularizers/constraints in the loss function. In that case, tuning the coefficients (which are hyper-parameters) becomes very hard, and often not doable. Hence, I doubt the effectiveness of the method in these cases. (I think this should be a very common situation in physics, as there are many conservation laws or other rules).

> [Minor] The paper claims “changing neural network architectures could potentially reduce the expressiveness of the neural network”. I am not convinced, I hope the authors provide some evidence for this claim.

---

> ### Author Response · Authors · 2023-02-21
> **Replies to Reviewer vbRE**
>
> > The idea of adding regularization to penalize unwanted aspects of learning is a very common trick in the machine learning community. Hence, the novelty is limited.
>
> While we acknowledge that adding a regularisation term is a common technique in machine learning, we want to highlight that in our paper, we use a physics-inspired regularisation term to address dynamical systems, which we believe is a novel approach. Furthermore, as TMLR prioritises technical accuracy over subjective significance, we do not consider novelty to be the primary concern.
>
> > The experiments shown in the paper are only very simple physics systems, which have very simple physics solutions. For a good evaluation of the proposed method, I expect it is tested on more complicated systems.
>
> The physical systems selected for our paper are consistent with those presented in the existing literature, such as HNN (Greydanus et al., 2019) and NSF (Chen et al., 2021) experiments, which include noisy image data.
>
> > I have doubts on the effectiveness of the proposed method. I noticed that all the experiments use super large constraint coefficients so that the enforced constraints are strictly obeyed. According to my personal experience, there is always a tradeoff between learning the system (corresponding to the loss without regularization) and enforcing the regularization. When the constraint coefficient is so large as in this paper, the machine learning models usually learn little to nothing meaningful. Hence, I suspect that the good performance in the presented experiments might be because the physics systems are so simple that they are very easy to learn. If this is the case, the proposed method may not work well for complicated systems.
>
> One approach to ensuring that the system satisfies the physical constraints is to apply a strong regularisation as proposed in our paper. With regularisation, there is some leeway in the training path to choose a route that deviates slightly from the physical constraints in order to achieve the minimum loss function. This is in contrast to other works, such as HNN and LNN, which impose constraints on the architecture from the outset. Architecture constraints strictly prohibit the training path from deviating from the physical constraints. We have tested our method with noisy image data, which is a more complex system, and our results have shown that the method performs well.
>
> > Another doubt I have is that: when there exist several inductive biases (for example several conservation laws), there should be multiple regularizers/constraints in the loss function. In that case, tuning the coefficients (which are hyper-parameters) becomes very hard, and often not doable. Hence, I doubt the effectiveness of the method in these cases. (I think this should be a very common situation in physics, as there are many conservation laws or other rules).
>
> We acknowledge that tuning the coefficients for multiple regularisation terms in the loss function can be challenging. However, in our paper, we concentrate on energy-conserving and dissipative systems that have an energy-based inductive bias, which entails that there is only one inductive bias in any given system. While other systems that involve multiple constraints are certainly interesting and could be a topic for future research, they are outside the scope of this paper.
>
> > [Minor] The paper claims “changing neural network architectures could potentially reduce the expressiveness of the neural network”. I am not convinced, I hope the authors provide some evidence for this claim.
>
> HNN is a simple example in which the network architecture is specifically created to ensure that the symplectic structure of the system is always preserved. However, the expressiveness of HNN is limited in the sense that the input data must be prepared in canonical form, which is usually difficult to obtain. Additionally, the HNN architecture cannot express the dynamics of dissipative systems.
>
> > I think the authors need to evaluate the method in more complicated physics systems, and to show it works well in these scenarios.
>
> We have addressed this comment above.
>
> > I hope the authors can show an efficient way of managing the scenarios of multiple inductive biases and constraints.
>
> We have addressed this comment above.
>
> > I would like the authors to also provide evidence for “it has been shown that to incorporate a specific inductive bias it takes a significant effort ……”
>
> We have addressed this comment above.
>
> > Provide evidence for the claim “changing neural network architectures could potentially reduce the expressiveness of the neural network”.
>
> We have addressed this comment above.

---

### Review · Reviewer_jz52 · 2023-02-10

**Summary Of Contributions:**

The authors propose to use a simple theoretically motivated regularisation term when modelling neural ODEs in order to conserve energy of the systems. This works with and without change of the coordinate systems for which the authors propose to learn invertible neural networks. They also provide a regularisation term to learn to model dissipative systems.

On a set of dynamical systems as well on a task which requires inferring states from pixel data, they test their method against related work and show promising results.

**Audience:**

Yes

**Broader Impact Concerns:**

-

**Claims And Evidence:**

Yes

**Requested Changes:**

1. Can you provide training curves of your method on the tasks?
2. Can you provide details on how the hyperparameters especially the regularization strength influences the behaviour?
3. It would be interesting to see the influence of the network depth and width on the performance/generalisation.
3. Why is the regularization strength so high? The loss function seem not correctly normalized? As far as explained you are using the squared-error loss, why not the MSE?
4. Can you please revisit the section about training instabilities of the NSF method? As far as I can see, the method does not suffer from instabilities (divergence) but just does not converge for some of the runs. I think it is a bit far fetched to say that the method suffers from instabilities. These slow training dynamics could e.g. be easily be due to improper tuning of the network initialization or optimizers. If you really want to make a point on this, please provide more details and tune all relevant hyperparameters.
5. What kind of network initialisation are you using?
6. Can you provide pseudo-code for your method?
7. It would be interesting to get a feeling of the computational overhead of the method / reg term. Are these Jacobians/Hessians expensive to compute?
8. Can you compare the training times (in sample efficiency) of your with related work?


**Strengths And Weaknesses:**

The paper is quite easy to understand and I like the simplicity of the proposed method which seems to perform well across the proposed tasks. I can not judge the novelty of the method, here i would like to hear the other reviewers opinions. The method feels like one of the first things to try.
Although I think the mathematical presentation is sufficient and clear, a few of experimental details would be important to show in the paper. Here, there are a few things that need clarification in order for acceptance in my opinion, see requested changes.

---

> ### Author Response · Authors · 2023-02-21
> **Replies to Reviewer jz52**
>
> > Can you provide training curves of your method on the tasks?
>
> Yes, we have updated the training curves in the appendix in the revised version of our paper, see Figure 9 in the appendix.
>
> > It would be interesting to see the influence of the network depth and width on the performance/generalisation.
>
> Our model is a simple Neural ODE with regularisation, so the usual influences of the network depth and width should be expected. In general, deeper networks are capable of learning more complex representations of the data, but are also more prone to overfitting. On the other hand, wider networks have more capacity to learn a large number of features and can reduce the risk of overfitting.
>
> > Can you provide details on how the hyperparameters especially the regularization strength influences the behaviour?
> > Why is the regularization strength so high? The loss function seem not correctly normalized? As far as explained you are using the squared-error loss, why not the MSE?
>
> The choice of regularisation strength is critical to ensure that the physical constraints are satisfied in the proposed method. A low regularisation strength could cause the neural network to prioritise accurate predictions over physical constraints, leading to potential overfitting, especially in the presence of noisy observations. In order to avoid this, a high regularisation strength is used to emphasise the importance of satisfying the physical constraints.
>
> As opposed to the reviewer’s suggestion of using squared-error loss, we are actually employing MSE as the metric for the experiments.
>
>
> > Can you please revisit the section about training instabilities of the NSF method? As far as I can see, the method does not suffer from instabilities (divergence) but just does not converge for some of the runs. I think it is a bit far fetched to say that the method suffers from instabilities. These slow training dynamics could e.g. be easily be due to improper tuning of the network initialization or optimizers. If you really want to make a point on this, please provide more details and tune all relevant hyperparameters.
>
> We recognise that the NSF training does not diverge. The word "instabilities" in this context refers to the fact that the training does not converge, i.e. the training loss plateaus before the optimum weights are found as shown in Figure 7 in the paper, although we acknowledge that it may have been an incorrect choice of word. We have replaced the term with "inconsistencies" in the updated version instead. We made an effort to address this inconsistency in the NSF training and discovered that using a smaller batch size helped alleviate the problem to some extent, but the issue persisted.
>
> > What kind of network initialisation are you using?
>
> The default network initialisation in PyTorch is used, i.e. He initialisation.
>
> > Can you provide pseudo-code for your method?
>
> The code has been made publicly available, and the link to the anonymised repository can be found in the paper.
>
> > It would be interesting to get a feeling of the computational overhead of the method / reg term. Are these Jacobians/Hessians expensive to compute?
>
> While computing the Jacobians and their eigenvalues can be computationally expensive, there are techniques to reduce the computational burden. For instance, the largest eigenvalues in Equation 8 can be computed efficiently using power iteration which only requires the Jacobian-vector product (i.e. forward derivative) that is available in modern deep learning frameworks.
>
> > Can you compare the training times (in sample efficiency) of your with related work?
>
> Task 1 provides a representative example of the training time for the various models. The times taken to train the NODE, OURS, HNN, LNN, and NSF models are 22, 29, 27, 53, and 34 minutes, respectively. NODE is the fastest method due to its simpler form. Our proposed method and HNN have similar training times, while LNN and NSF are slower to train.

---

### Decision · Action_Editors · 2023-04-05

**Recommendation:** Accept as is

**Comment:**

This paper proposes adding a regularization term to the loss function that can capture physical constraints such as energy conservation. It evaluates the method on several simple benchmark problems and compares the results with some natural baselines.

It was noted in the discussion that adding a regularization penalty is such a common method in machine learning that the results here may not interest the community. Of course, sometimes well-known techniques can lead to important insights if applied in new ways. In this case, the form of the regularizer is perhaps not particularly surprising, but nonetheless there is value in demonstrating its efficacy.

It was also noted that the applications were restricted to simple physical systems, and it was unclear whether the technique would generalize to more complex scenarios. The examples investigated here are indeed fairly simple, but the results are compelling. Furthermore,  no matter where the analysis stopped, there would always be more complex systems out there that were not investigated; it is thus reasonable and natural to limit initial investigations to the simplest settings.

All told, while there are ways the manuscript might be improved in order to further its impact, as it stands there is sufficient value to merit publication.

**Audience:**

Certainly Neural ODEs and modeling dynamical constraints are topics that interest parts of the TMLR community. While the methods presented here are simple and the results unsurprising, some individuals may find the conclusions useful in guiding their research and development.

**Claims And Evidence:**

The reviewers mostly agreed that the claims were well-supported. One concern was that the set of evaluations was limited to simple systems. I agree that a broader set of evaluations that included more complicated systems would improve the paper, but I do not think such extensions are strictly necessary to support the main claims.